**Modeled source apportionment of black carbon particles coated with a light-scattering shell**
Aki Virkkula
Finnish Meteorological Institute
Helsinki, Finland
**Abstract**
The Aethalometer model has been used widely for estimating the contributions of fossil fuel emissions
and biomass burning to equivalent black carbon (eBC). The calculation is based on measured absorption
Ångström exponents ($\alpha_{abs}$). The interpretation of $\alpha_{abs}$ is ambiguous since it is well-known that it not
only depends on the dominant absorber but also on the size and internal structure of the particles, core
size and shell thickness. In this work the uncertainties of the Aethalometer-model-derived apparent
fractions of absorption by eBC from fossil fuel and biomass burning are evaluated with a core-shell Mie
model. Biomass-burning fractions (BB(%)) were calculated for pure and coated single BC particles, for
lognormal unimodal and bimodal size distributions of BC cores coated with ammonium sulfate, a
scattering-only material. BB(%) was very seldom 0% even though BC was the only absorbing material in
the simulations. The shape of size distribution plays an important role. Narrow size distributions result
in higher $\alpha_{abs}$ and BB(%) values than wide size distributions. The sensitivity of $\alpha_{abs}$ and BB(%) to
variations in shell volume fractions is the highest for accumulation mode particles. This is important
because that is where the largest aerosol mass is. For the interpretation of absorption Ångström
exponents it would be very good to measure BC size distributions and shell thicknesses together with
the wavelength dependency of absorption.
**1. Introduction**
Incomplete combustion of organic fuels results in emission of light-absorbing carbon (LAC) particles
that contain both black carbon (BC) and brown carbon (BrC).  BrC is light-absorbing organic matter in
atmospheric aerosols of various origins e.g., soil humics, humic-like substances (HULIS), tarry materials
from combustion, bioaerosols (Andreae and Gelenscer, 2006; Laskin et al., 2015). BrC can significantly
absorb solar radiation in the ultraviolet–visible (uv–vis) wavelength range (λ ≈ 300 − 800 nm). The

radiative effects of BC and BrC vary in time during atmospheric aging. For many combustion sources the absorbance in fresh emission is almost completely caused by BC particles but during atmospheric transport they often get coated with some light-scattering compounds, for instance ammonium sulfate or light-absorbing organic carbon, BrC. For some sources (e.g. biomass burning) BrC may contribute substantially to light-absorption already in the directly emitted aerosols and either increase or decrease during aging. Thus, BrC is highly time-dependent as it's composition and absorption properties change during atmospheric oxidation processes (Laskin et al., 2015).

The absorption coefficient $\sigma_{ap}$ is approximately proportional to the power function $\lambda^{-\alpha_{abs}}$ where $\lambda$ is the wavelength and $\alpha_{abs}$ is the absorption Ångström exponent. $\alpha_{abs}$ is generally used to distinguish aerosol types: for pure BC particles $\alpha_{abs} \approx 1$ while other light absorbing aerosols (BrC, soil dust) it is clearly > 1 (e.g., Kirchstetter et al., 2004; Bond and Bergstrom, 2006; Bergstrom et al., 2007; Moosmüller et al, 2011; Kirchstetter and Thatcher, 2012; Lack et al., 2012; Bond et al., 2013; Saleh et al., 2013; Laskin et al., 2015; Valenzuela et al., 2015; Devi et al., 2016). The method has been used not only for in situ absorption measurements but also for interpreting absorption coefficients retrieved from remote sensing measurements, such as the AERONET (e.g., Russell et al., 2010; Arola et al., 2011; Chung et al., 2012; Cazorla et al., 2013; Feng et al., 2013; Schuster et al., 2016; Wang et al., 2016).

One of the instruments used for measuring black carbon concentrations is the Aethalometer that collects aerosol on a filter tape, measures changes in light attenuation in the wavelength range of 370 – 950 nm and calculates the equivalent black carbon (eBC) concentrations. The data are used also to calculate $\alpha_{abs}$ and to estimate the contributions of fossil fuel emissions and biomass burning to eBC. The Aethalometer model (Sandradewi et al., 2008a) is probably the most widely-used method for this and it is even calculated automatically in the new Aethalometer model AE33. It is there assumed that the absorption Ångström exponents are $\alpha_{ff} = 1$ and $\alpha_{bb} = 2$ for eBC from fossil fuel and biomass burning, respectively. These are the default settings in the AE33, but also different $\alpha_{ff}$ and $\alpha_{bb}$ values have been used (Sandradewi et al., 2008b; Herich et al., 2011; Fuller et al., 2014; Harrison et al., 2013; Healy et al., 2017; Zotter et al., 2017; Helin et al., 2018)

The interpretation of $\alpha_{abs}$ is ambiguous since it not only depends on the dominant absorber but also on
the size and internal structure of the particles, core size and shell thickness. For instance, for pure BC
particles, $\alpha_{abs}$ may be < 1 and BC particles coated with non-absorbing material may have $\alpha_{abs}$ in the
range from <1 to ~1.7 (e.g., Gyawali et al., 2009; Lack and Cappa, 2010; Lack and Langridge, 2013;
Schuster et al., 2016; Liu et al., 2018; Chylek et al., 2019; Zhang et al., 2020). The present paper may be
considered as an extension to the above-mentioned analyses since they did not explicitly analyze the
effects on the Aethalometer model.
The aim of this study is to estimate uncertainties of the Aethalometer-model-derived fractions of
absorption by eBC from fossil fuel and biomass burning when spherical BC cores are coated by some
non-absorbing material. To state this more clearly, it is assumed that there is only one type of BC
particles that can be called fossil fuel BC in the Aethalometer model terminology. Consequently, any
deviations from biomass-burning fraction of BB% = 0 indicate uncertainty in the source appointment.
Biomass-burning fractions were calculated for pure and coated single particles, for lognormal unimodal
and bimodal size distributions. The work is based on modeling only, no measurement data are used.
**2. Methods**
The BC cores were assumed to be coated with an ammonium sulfate shell by using two approaches. It
was assumed 1) that the shell thickness the same for all particles in a size distribution (Fig. 1a) and 2)
that the core volume fraction is the same for all particles in a size distribution (Fig. 1b). The core volume
fraction was calculated from

$$f_c = \frac{V_{core}}{V_p} = \left(\frac{D_{core}}{D_p}\right)^3 = \left(\frac{D_{core}}{D_{core}+2s}\right)^3 \tag{1}$$

where $V_p$ is the particle volume, $V_{core}$ is the BC core volume, $D_p$ is the particle diameter (= $D_{core}$ + 2s),
$D_{core}$ is the BC core diameter and s the shell thickness. The shell volume fraction was then calculated
from $f_s = 1 - f_c$. The ratio of the coated particle diameter to the core diameter  is an often used metric
for presenting the coating of particles. $R$, $f_c$ and $f_s$ can be calculated from each other as

$$R = \frac{D_p}{D_{core}} = \left(\frac{1}{f_c}\right)^{1/3} = \left(\frac{1}{1-f_s}\right)^{1/3} \tag{2}$$

The number-weighted $D_p$-to-$D_{core}$ ratio is calculated from

$$R_{N(D_p)} = \frac{\sum N_i R_i}{N_{tot}} = \frac{\sum N_i \left( D_{p,i} / D_{core,i} \right)}{N_{tot}} \tag{3}$$

where $N_i$ and $R_i$ are the number concentration and $D_p$-to-$D_{core}$ ratio of the particle diameter $D_{p,i}$, respectively. If $f_s$ is independent of particle size – which is the assumption used in some of the simulations below – equation (3) simplifies to $R_{N(D_p)} = R$.

Lognormal size distributions $n(D_p, D_g, \sigma_g)$ were generated where $D_p$ is the particle diameter, $D_g$ is the geometric mean diameter and $\sigma_g$ the geometric standard deviation. The $D_p$ range was 3 nm – 10 µm. For the unimodal size distributions $D_g$ range was 50 nm – 1 µm and $\sigma_g$ was given three values 1.4, 1.6, 1.8 (Fig. 1c and 1d). Also bimodal size distributions were generated. For the small-particle mode the geometric mean diameter $D_{g1}$ range was 50 – 100 nm, and the large-particle mode $D_{g2}$ range was 100 – 500 nm. In addition to varying the geometric mean diameters also the ratios of the number of particles in the two modes were varied. Two cases were used for this: 1) $N_1 = 10N_2$, $\sigma_{g1} = 1.4$, $\sigma_{g2} = 1.6$ (Fig. 1e) and 2) $N_1 = N_2$, $\sigma_{g1} = 1.6$, $\sigma_{g2} = 1.6$ (Fig. 1f).

Absorption coefficients were calculated from

$$\sigma_{ap}(\lambda) = \int Q_a(\lambda, D_p, m_{core}, m_{shell}, s) \frac{\pi}{4} D_p^2 n(D_p) dD_p \tag{4}$$

where $Q_a$ is the absorption efficiency that is a function of the wavelength $\lambda$, $D_p$, the complex refractive indices of the core and shell, $m_{core}$ and $m_{shell}$, respectively, and the shell thickness s. $Q_a$ was calculated using the N-Mie Fortran code that is based on a recursive algorithm of Wu and Wang (1991). The code calculates the extinction, scattering and absorption efficiency factors for n-layered spheres. The complex refractive indices were $m_{core}$ = 1.85 + 0.71i (BC as in Lack and Cappa, 2010) and $m_{shell}$ = 1.52+0i (ammonium sulfate) for the core and shell, respectively. Absorption coefficients were calculated for the Aethalometer wavelengths $\lambda$ = 470 nm and 950 nm and $\alpha_{abs}$ was calculated from

$$\alpha_{abs}(470/950) = -\frac{\ln\left( \sigma_{ap}(\lambda = 470nm) / \sigma_{ap}(\lambda = 950nm) \right)}{\ln\left( 470/950 \right)} \tag{5}$$

The wavelengths 470 nm and 950 nm were used as they are used also in the AE33 automatic source apportionment. In analyses of aerosol optical depth data from the AERONET network $\alpha_{abs}$ is often

calculated for the wavelength pair 440 nm and 870 nm (Russell et al., 2010; Schuster et al., 2016). To evaluate the applicability of the simulations of the present work to AERONET data analyses $\sigma_{ap}$ was calculated also for these wavelengths and the respective $\alpha_{abs}$ was calculated from them. There are size-dependent differences between $\alpha_{abs}(470/950)$ and $\alpha_{abs}(440/870)$ but they are not big, see the supplement, Figs. S1 and S2, so it may safely be concluded that the results to be presented below are valid also for the AERONET data.

For the absorption due to particles from wood burning or biomass burning Zotter et al. (2017) give the equation

$$\sigma_{ap,bb}(\lambda_2) = \frac{\sigma_{ap}(\lambda_1) - \sigma_{ap}(\lambda_2)\left(\dfrac{\lambda_1}{\lambda_2}\right)^{-\alpha_{ff}}}{\left(\dfrac{\lambda_1}{\lambda_2}\right)^{-\alpha_{bb}} - \left(\dfrac{\lambda_1}{\lambda_2}\right)^{-\alpha_{ff}}} \tag{6}$$

where $\alpha_{ff}$ and $\alpha_{bb}$ are the $\alpha_{abs}$ of fossil fuel and biomass burning BC in the Aethalometer model. Noting that $\sigma_{ap}(\lambda_1) = \sigma_{ap}(\lambda_2)(\lambda_1/\lambda_2)^{-\alpha_{abs}}$ the fraction of absorption due to biomass burning is

$$BB(\%) = 100\% \frac{\sigma_{ap,bb}(\lambda_2)}{\sigma_{ap}(\lambda_2)} = 100\% \frac{\left(\dfrac{\lambda_1}{\lambda_2}\right)^{-\alpha_{abs}} - \left(\dfrac{\lambda_1}{\lambda_2}\right)^{-\alpha_{ff}}}{\left(\dfrac{\lambda_1}{\lambda_2}\right)^{-\alpha_{bb}} - \left(\dfrac{\lambda_1}{\lambda_2}\right)^{-\alpha_{ff}}} \tag{7}$$

so that BB% depends on the Ångström exponents $\alpha_{abs}$, $\alpha_{ff}$ and $\alpha_{bb}$. Two settings for the constants were used, the one presented in the AE33 manual: $\alpha_{ff}$ = 1 and $\alpha_{bb}$ =2, and the one presented by Zotter et al. (2017): $\alpha_{ff}$ = 0.9 and $\alpha_{bb}$ = 1.68.

## 3. Results and discussion

### 3.1 Single particles

The absorption Ångström exponent $\alpha_{abs}$ and the fraction of biomass-burning BC for single coated particles are shown in Fig. 2. The dashed lines in Figs. 2a, 2c, and 2e show the core diameter $D_{core}$ of particles that have the same diameter $D_p$ at all shell thicknesses. In Figs. 2b, 2d and 2f the dashed lines show the particle diameter $D_p$ and $f_s$ of particles that have the same $D_{core}$ at all shell volume fractions $f_s$

5

in the range $f_s \leq 99$ %. The dependence of $\alpha_{abs}$ on core and shell is presented twice. This is apparently
superfluous but they are visualizations that complement each other.
The first approach (Figs. 2a, 2c, and 2e) shows that when $D_{core} < \sim150$ nm and $s > \sim25 - 50$ nm the
absorption Ångström exponent $\alpha_{abs} > 1.4$. The respective BB fractions are larger than about 40% or 60%
for the Aethalometer model parameters of $\alpha_{ff} = 1$, $\alpha_{bb} = 2$ (pair 1) and $\alpha_{ff} = 0.9$, $\alpha_{bb} = 1.68$ (pair 2),
respectively. Fig. 2a also shows that for $D_{core} < \sim100$ nm there are two maxima of the $\alpha_{abs}$ when the
shell grows thicker. In the second maximum $\alpha_{abs} > \sim1.6$. As a result the BB fractions would be > 60%
and even > 100% for the two Aethalometer model parameter pairs. When $D_{core}$ is in the range of $\sim$170-
200 nm $\alpha_{abs} \approx 1$ and $\alpha_{abs}$ decreases with a growing s. For larger core diameters the absorption Ångström
exponent is even smaller. When $D_{core} > 200$ nm $\alpha_{abs} < 1$, and even negative for $D_{core} > \sim360$ nm. Further,
when $D_{core} > 200$ nm, $\alpha_{abs}$ does not grow essentially at all as a function of s.
The visualization of $\alpha_{abs}$ as a function of shell volume fraction $f_s$ and particle full diameter $D_p$ (Fig. 2b)
shows some other features. When $D_p < 50$ nm, $\alpha_{abs}$ varies in the range of 1.0 - 1.1 and it does not depend
on $f_s$ but in the $D_p$ range of about $100 - 300$ nm $\alpha_{abs}$ depends strongly on $f_s$. When $D_p \approx 500$ nm $\alpha_{abs} <$
1 for almost all shell volume fractions, up to $f_s \sim99$%. For larger particles $\alpha_{abs}$ is close to 0 at all shell
volume fractions.
The visualization also shows that the $\alpha_{abs}$ value of 1, usually considered as indication of BC, is not a
result of an unambiguous $D_{core}$-s (Fig. 2a) or $D_p$-$f_s$ (Fig. 2b) combination.
**3.2 Unimodal BC core size distributions, same coating thickness for all sizes**
For single particles $\alpha_{abs}$ depends clearly both the core size and the shell thickness. However, in real
atmospheric studies the wavelength dependency of absorption by particle size distributions are
measured. Here these were first modeled by assuming that pure BC particle size distributions get
coated with ammonium sulphate layers so that the shell thickness is independent of particle size as
visualized in Fig. 1a. For example, the shell thickness on a 50 nm BC particle would be the same as on a
200 nm particle which means the shell volume fractions are not the same. The BC core geometric mean
diameter ($D_{g,core}$) was varied from 50 to 200 nm at 10 nm intervals. The geometric standard deviations
of the size distributions were $\sigma_g$ = 1.4, $\sigma_g$ = 1.6, and $\sigma_g$ = 1.8 representing narrow, average and wide
size distributions. The shell thickness s varied from 0 to 250 nm at 1 nm intervals. Absorption coefficient
and subsequently $\alpha_{abs}$ was calculated for the full size distribution 3 – 2500 nm.
The results are first shown as a function of $D_{g,core}$ and shell thickness s for the three size distribution
widths (Fig. 3). There are both similarities and differences compared with the corresponding
relationships of single particles (Fig. 2). For example, for single particles $\alpha_{abs} \approx 1$ at $D_{core} \approx 180$ nm for
shell thicknesses $s \approx 0 - 70$ nm as shown by the almost vertical $\alpha_{abs} = 1$ isoline in Fig. 2a  whereas for
the size distributions with $\sigma_{g,core} > 1$  the respective isoline is a strong function of both s and $\sigma_{g,core}$ (Fig.
3a).  At all widths of the size distribution $\alpha_{abs}$ increases with increasing shell thickness and then starts
decreasing. For small core sizes ($D_{g,core} <$ ~80 nm)  $\alpha_{abs}$ has also a second maximum when the size
distribution is narrow.  The width of the size distribution has a clear effect on the $\alpha_{abs}$: for all core sizes
and shell thicknesses $\alpha_{abs}$ decreases with increasing $\sigma_{g,core}$.
Both for single particles and size distributions the first maximum of $\alpha_{abs}$ is the smaller the larger the
$D_{g,core}$ and $\sigma_{g,core}$ are (Fig. 4a). The first maximum is reached at shell thickness $s \approx 70 \pm 5$ nm for all size
distribution widths although for single particles the variability of the shell thickness corresponding to
the first maximum is larger (Fig. 4b). The first maximum $\alpha_{abs}$  results in apparent BB fractions of up to
~100% for single particles and in the range from 0 to ~80% for the size distributions and again the BB(%)
is the smaller the larger the $D_{g,core}$ and $\sigma_g$ are (Fig. 4c and d).
This approach is further followed by plotting the parameters as a function of shell thickness for three
different BC core diameters, 50 nm, 70 nm, and 90 nm of single particles and core size distributions
with the geometric standard deviations of  $\sigma_{g,core}$ = 1.4, 1.6, and 1.8 (Fig. 5). This analysis can be
considered as a description of what may happen to the size distribution, $\alpha_{abs}$ and the apparent BB(%)
during condensational growth on fresh small BC cores if the growing shell thickness were independent
of the core diameter, even if this is unrealistic. The shell volume fraction $f_s$ increases to > 99.9% when
the shell thickness grows from s = 0 nm to 250 nm on single 50 nm particles but to lower fractions for
the wider size distributions and larger core sizes so that for $D_{g,core}$ = 90 m and $\sigma_g$ = 1.8 $f_s \approx$ 98% even
with s = 250 nm (Fig. 5a). The geometric mean diameter $D_g$ of the whole size distribution grows to ~600
nm when the shell thickness grows to 250 nm, minimal differences between the original widths (Fig.
5b). The width of the size distribution, i.e., $\sigma_g$ decreases fast to < 1.2 (Fig. 4c). Such values correspond
to very  narrow size distributions, not really observed in the real atmosphere.
The number-weighted $D_p$-to-$D_{core}$ ratio $R_{N(Dp)}$, Eq. (3), was calculated for the size range 90-600 nm to
present the numbers comparable with papers that present shell-to-core ratios of refractory BC (rBC)
obtained from SP2 measurements. For instance, Kondo et al. (2011) measured urban air of Tokyo and
obtained the median R = 1.1 with a range up to about 1.3, the mean $D_g$ = 64 ± 6 nm, and $\sigma_g$ = 1.66 ±
0.12. Moteki et al. (2007) conducted SP2 measurements in an aircraft in urban plumes on the Japanese
coast. They fitted the data with lognormal size distributions with mass median diameters (MMD) of 190
and 210 nm and $\sigma_g$ of 1.55 and 1.45 for fresh and aged rBC, respectively. The fresh rBC was  thinly
coated with R < 2 and the aged rBC thickly coated with R ~ 2.  The MMD and $\sigma_g$ values yield  $D_g$ = 107
nm and 139 nm.  Shiraiwa et al. (2008) measured the mixing state and size distribution of BC aerosol
with an SP2 at a remote island (Fukue) in Japan. They observed that the BC  number median diameters
were in the range of 120–140 nm in every air mass type and the median shell/core diameter ratio (R)
in different air masses varied in the range of 1.2 – 1.6. However, they also observed that the fraction of
R values in the range 2 – 3.5 was not negligible either (Fig. 9 of Shiraiwa et al., 2008). Such values
correspond to the range where $\alpha_{abs}$ first grows to >1.6 for the  narrow ($\sigma_{g,core}$ = 1.4) BC core size
distribution that has the smallest geometric mean size ($D_{g,core}$ = 50 nm)  but to lower values for the
wider size distributions that have larger $D_{g,core}$ (Fig. 5c and 5d). The first maximum is reached at shell
thicknesses of s $\approx$ 70 nm that corresponds to R $\approx$ 2 and shell volume fractions of $f_s \approx$ 90 ± 8% (Fig. 5b).
Schwarz et a. (2008) reported statistics of rBC mass size distributions in urban aerosol: $f_s$ = 9 ±  6%, s =
20 ± 10 nm, MMD = 170 nm, and $\sigma_g$ of 1.71 which yields $D_g$ = 72 nm; in biomass burning emissions: $f_s$ =
70 ± 9 %, s = 65 ± 12 nm, MMD = 210 nm, $\sigma_g$ =1.43 which yields $D_g$ = 143 nm and in background
continental aerosol: $f_s$ = 46 ± 3%, s = 48 ± 14 nm, MMD = 210 nm, 1.55 which yields $D_g$ = 118 nm.
The referenced studies show that the s, R, and $f_s$ values are in the range observed in ambient
measurements studies. What is not realistic in atmospheric aerosol is the width of the size distribution,
which soon decreases to $\sigma_g$ < 1.2 (Fig. 5c).
After reaching the first maximum $\alpha_{abs}$ decreases and for single particles and the narrowest core size
distributions starts again growing and reaches a second maximum at shell thicknesses of s ≈ 170 nm
that corresponds to R > 4 and $f_s$ > 98%. Such s and R values are not in the range observed in the above-
mentioned studies, nor are the low geometric standard deviations of $\sigma_g$ < 1.1 realistic so the second
maximum can be considered as a theoretical value only. For size distributions with $D_{g,core}$ > 70 nm there
is no second maximum of $\alpha_{abs}$.
As $\alpha_{abs}$ increases and decreases it is clear that this applies to BB(%) as well (Fig. 5d-e). For the smallest
core sizes ($D_{g,core}$ = 50 nm) and the narrowest size distributions ($\sigma_{g,core}$ = 1.4) the first maximum BB(%)
may be as high as ~90% when the values of $\alpha_{ff}$ = 0.9, $\alpha_{bb}$ = 1.68 are used in Eq. (7) but lower, ~50%
when the values of $\alpha_{ff}$ = 1, $\alpha_{bb}$ = 2 are used. For the wider core size distributions the BB(%) fractions are
lower. For the widest core size distributions ($\sigma_{g,core}$ = 1.8) clearly positive BB(%) values are reached only
for the smallest core sizes.
Fig. 5 can also be considered as a proxy for a time series of the development of $\alpha_{abs}$ and derived BB(%)
after emission of BC particles and their growth by condensation of nonabsorbing compounds. Similar
development – $\alpha_{abs}$ increase to > 1.3 and decrease to < 0.9 during a several-day-long pollution episode
during which the $D_g$ of the whole size distribution grew possibly by condensation – was observed at
SORPES in Nanjing, China (Fig. 9 of Shen et al., 2018). There was no SP2 available for the core-shell
structure measurements in that study so it cannot really be proven that the observed $\alpha_{abs}$ development
was due to condensational growth even though it seems a good explanation and is qualitatively in line
with Fig. 5.

**3.3 Unimodal size distributions with the same BC core fraction for all sizes**

The second approach is to assume that the BC core fraction – or equivalently the shell volume fraction – is the same for all sizes which means that the shell thickness increases with size as was visualized in Fig. 1b. This can be considered to be a result of aging of BC by not only condensational growth but also by cloud processing. The latter would lead to thick shells on particles activated into cloud droplets that would absorb for instance $SO_2$ and $NH_3$ and that would not rain but get later back into the aerosol phase by evaporation of cloud water. The constant volume fraction is not realistic but neither is the constant shell thickness. Both can be considered to be approximations.

In this approach the geometric standard deviations of the whole size distributions were set to $\sigma_g$ = 1.4, 1.6 and 1.8 and the shell volume fractions $f_s$ to vary from 0% to 99%. The resulting $\alpha_{abs}$ and BB(%) are presented as a function of $D_g$, $f_s$ and $\sigma_g$ (Fig. 6). They are comparable with the analogous plots for single particles, i.e., $\sigma_g$ = 1.0 (Fig. 2b, 2d, and 2f). Note that from Eq. (2) it follows that the assumption of a constant $f_s$ means that also the $D_p$-to-$D_{core}$ ratio R is constant and that the $f_s$ range of 0 to 99% corresponds to the R range of 1 to 4.6. Figure 6 therefore has two y axes, one showing the $f_s$ and the other the corresponding R values.

Several observations can be made from Fig. 6. One of them is that the isoline of $\alpha_{abs}$ = 1 grows with growing $D_g$ for each of the size distribution widths ($\sigma_g$) but decreases with growing $\sigma_g$. Another is that the wider the size distribution is, the lower are the $\alpha_{abs}$ and BB(%) at any given shell volume fraction. The third one is that for all three widths $\alpha_{abs}$ and BB(%) grows when $f_s$ grows but that the growth is not uniformly distributed over the $f_s$ vs. $D_g$ space.

The last observation leads to calculations of size-dependent sensitivities of $\alpha_{abs}$ to variations in $f_s$. The sensitivity was calculated as $d\alpha_{abs}/df_s$ and its unit is $\%^{-1}$. Fig. 7a shows the sensitivities in the whole $f_s$ range of 1 - 99% as a function of $D_g$ for the three size distribution widths. The sensitivity depends clearly on both $D_g$ and $\sigma_g$ of the size distribution and it also varies with $f_s$. It is very clear that $\alpha_{abs}$ is most sensitive to $f_s$ variations when $D_g$ of the size distribution is in the accumulation mode sizes of 100 – 200

nm. The sensitivity grows fairly steadily with growing $f_s$ until for $f_s > 90\%$ – which equals $R > 2$ – it
increases very strongly.
Another step for visualizing the sensitivities was taken by calculating size-dependent average
sensitivities of $\alpha_{abs}$ and BB(%) in three $f_s$ ranges: $f_s = 0 - 50\%$, $50 - 90\%$ and $90 - 99\%$ (Fig. 7b and 7c).
According to Eq. (2) the $f_s$ ranges correspond to the R ranges of $1 - 1.3$, $1.3 - 2.2$ and $2.2 - 4.6$. The lines
in Fig. 7b and 7c can be used for a rough estimate on a possible effect on $\alpha_{abs}$ and BB(%). For instance,
if $D_g \approx 140$ nm, $\sigma_g = 1.4$, and $f_s \approx 50 - 90\%$ , an increase of $f_s$ from 50% to 51% leads to an $\alpha_{abs}$ increase
of ~0.01 and consequently to a BB(%) increase of ~1% when Aethalometer model constants of $\alpha_{ff} = 0.9$,
$\alpha_{bb} = 1.68$ are used.
**3.4 Bimodal size distributions with the same BC core fraction for all sizes in the mode**
Finally, bimodal size distributions are examined briefly. The size distributions consist of two externally
mixed modes that have different shell volume fractions. In both modes the shell volume fractions are
size-independent as in Fig. 1b.  Mode 1 is an Aitken mode with the geometric mean diameter $D_{g1}$ in the
range $50 - 100$ nm. There are two different settings for the Aitken mode: in the first case its number
concentration is 10 times larger than that of the accumulation mode, i.e., $N_1 = 10N_2$, it consists of almost
pure fresh BC particles with $f_{s1} = 5\%$ $(R \approx 1.02)$ and it is narrow, $\sigma_{g1} = 1.4$. In the second setting the
number concentrations of the Aitken and accumulation mode are equal ($N_1 = N_2$), the Aitken mode is
aged so that $f_{s1} = 50\%$ $(R \approx 1.3)$ and it is wider, $\sigma_{g1} = 1.6$. Mode 2 is an accumulation mode with the
geometric mean diameter $D_{g2}$ in the range $100 - 500$ nm, $\sigma_{g2} = 1.6$ and it is very aged, with $f_{s2} = 98\%$ $(R$
$\approx 3.7)$. The accumulation mode could be the result of cloud processing  as explained above.
The results show that $\alpha_{abs}$ is more sensitive to variations of the  accumulation mode than of the Aitken
mode (Fig. 8a). For instance, if $D_{g2} < 250$ nm, $\alpha_{abs} > 1$ at all $D_{g,1}$ values. Also, if $D_{g,1} = 60$ nm and $D_{g2}$ varies
in the whole range of $100 - 500$ nm, $\alpha_{abs}$ varies in the range of ~ $0.4 - 1.3$. When the Aitken mode
dominates the number concentration ($N_1 = 10N_2$) with the fresh BC particles the maximum $\alpha_{abs} \approx 1.2$ at
$D_{g1} \approx 60$ nm and $D_{g2} \approx 140$ nm is smaller than when the two modes have equal amount of particles. In
the latter case the maximum $\alpha_{abs} > 1.3$. When the Aitken mode with $f_s = 5\%$ dominates the number

concentration the whole size distribution moves to the region that is less sensitive to $f_s$ variations as discussed above in section 3.3. It is worth noting also that the maximum $\alpha_{abs}$ and BB(%) values (Fig. 8b and 8c) are smaller than derived from the unimodal size distributions (section 3.3).

**4. Summary and conclusions**

The purpose of this study is not to claim that all Aethalometer model results are wrong but that they have higher uncertainties than have been discussed in the literature. It is clear that there are BrC particles that have absorption Ångström exponents clearly larger than one, as shown in a very large number of publications. However, the size of light-absorbing particles and their coating even by purely scattering material affect clearly the wavelength dependence of absorption and thus have the potential to affect the Aethalometer model results. Since the wavelength dependency is used for source apportionment these effects have the potential to result in tens of percent too high or low contributions of wood-burning or fossil fuel emissions.

There are some important results. In the modeling $\alpha_{abs}$ equals 1 or 0.9 in very rare cases and thus BB(%) was very seldom 0% even though one type of BC was the only absorbing material in the simulations. The shape of size distribution plays an important role. Narrow size distributions result in higher $\alpha_{abs}$ and BB(%) values than wide size distributions. The sensitivity of $\alpha_{abs}$ and BB(%) to variations in shell volume fractions is the highest for accumulation mode particles. This is important because that is where the largest aerosol mass is.

The goal of the paper was not to find out whether some pair of $\alpha_{ff}$ and $\alpha_{bb}$ is better than the other. Two well-known $\alpha_{ff}$ and $\alpha_{bb}$ pairs were used and shown how large the uncertainties may become just for these two pairs even if BC particles were coated by purely scattering material. The goal was not at all to find a good pair. On the contrary, the study shows that no constant values are good since in the real atmosphere BC particle size distributions are not constant, neither their mean diameter nor the coating of the particles. They all vary dynamically in the atmosphere. The study shows that any constant values will undoubtedly lead to large uncertainties of both the BB and FF fractions if no information on the size of the core or the thickness of the shell is available, even if purely scattering material is coating BC

cores. As a conclusion, for the interpretation of absorption Ångström exponents it would be very important to measure BC size distributions and shell thicknesses together with the wavelength dependency of absorption.

There are obvious limitations in this study. A core-shell Mie model was used only so the work is limited to spherical particles. Fresh BC particles are usually agglomerates. There are studies that show that during aging processes these agglomerate may collapse and become closer to spherical particles so Mie modeling probably agrees better for aged than fresh BC particles. Further, even if particles were spherical how well can they be modeled with a Mie code when they are collected on filters? Or does light absorption then follow the spectral absorbance of the bulk materials?

This question could in principle be answered by generating spherical BC particles, coating them in an aging chamber with some non-absorbing material, for instance ammonium sulfate, and measuring both light absorption at multiple wavelengths with an Aethalometer and BC core size distributions and shell thicknesses with an SP2. If $\alpha_{abs}$ increases up to some maximum value as a function of shell thickness and then starts decreasing like in the simulations above, then the process agrees with the growth of a size-independent coating. Or if $\alpha_{abs}$ increases steadily then it suggests that the growth is size-dependent and possibly with a size-independent shell volume fraction growth rate. If these are observed then the uncertainties discussed in this work should be taken seriously.

On the other hand, if none of these effects were observed and the absorption Ångström exponents of the collected particles were $\approx 1$ regardless of core size and shell thickness it would be safe to say that the Aethalometer measures the absorption spectra of the bulk materials and that the Aethalometer model yields correct results. Probably the truth is somewhere between these extremes: when the filter tape is still relatively clean the particles can be modeled even with a Mie code and for heavily-load filters $\alpha_{abs}$ is that of bulk material. Also this could and should be tested experimentally.

*Acknowledgements* This work was supported by the Academy of Finland via project NABCEA (grant no. 296302) and by Business Finland via project BC Footprint (grant nr. 528/31/2019).

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

1 **Tables**

3 Table 1. Nomenclature

| Symbol | Definition | Unit | Equation |
|---|---|---|---|
| $D_p$ | Particle diameter | nm | (1) |
| $D_{core}$ | Diameter of the BC core particle | nm | (1) |
| $D_g$ | Geometric mean diameter of a size distribution | nm | |
| $D_{g,core}$ | Geometric mean diameter of the BC core size distribution | nm | |
| $D_{g1}$ | $D_g$ of the first mode of a bimodal size distribution | nm | |
| $D_{g2}$ | $D_g$ of the second mode of a bimodal size distribution | nm | |
| $\sigma_g$ | Geometric standard deviation of a size distribution | - | |
| $\sigma_{g,core}$ | Geometric standard deviation of the BC core size distribution | - | |
| $\sigma_{g1}$ | $\sigma_g$ of the first mode of a bimodal size distribution | - | |
| $\sigma_{g2}$ | $\sigma_g$ of the second mode of a bimodal size distribution | - | |
| $n(D_p,D_g,\sigma_g)$ | Lognormal particle number size distribution | cm$^{-3}$ | |
| $N_1$ | Number concentration of the first mode of a bimodal size distribution | cm$^{-3}$ | |
| $N_2$ | Number concentration of the second mode of a bimodal size distribution | cm$^{-3}$ | |
| $N_i$ | Number concentration of particle size $D_{p,i}$ | cm$^{-3}$ | (3) |
| $V_p$ | Particle volume | m$^3$ | (1) |
| $V_{core}$ | Volume of the BC core | m$^3$ | (1) |
| $f_c$ | Core volume fraction | - | (1) |
| $f_s$ | Shell volume fraction | - | |
| $s$ | Shell thickness | nm | (1) |
| $R$ | Ratio of the particle diameter to the BC core diameter ($D_p$-to-$D_{core}$ ratio) | - | (2) |
| $R_{N(Dp)}$ | Number-weighted $D_p$-to-$D_{core}$ ratio | - | (3) |
| $R_i$ | $D_p$-to-$D_{core}$ ratio of the particle diameter $D_{p,i}$ | - | (3) |
| $\sigma_{ap}(\lambda)$ | Absorption coefficient at the wavelength $\lambda$ | Mm$^{-1}$ | (4) |
| $\sigma_{ap,bb}(\lambda)$ | Absorption coefficient of particles from biomass burning at the wavelength $\lambda$ | Mm$^{-1}$ | (6) |
| $Q_a$ | Absorption efficiency | - | (4) |
| $m_{core}$ | Complex refractive index of the BC core | - | (4) |
| $m_{shell}$ | Complex refractive index of the shell | - | (4) |
| $\alpha_{abs}$ | Absorption Ångström exponent | - | |
| $\alpha_{abs}(\lambda_1/\lambda_2)$ | Absorption Ångström exponent for the wavelength pair $\lambda_1$, $\lambda_2$ | - | (5) |
| $\alpha_{ff}$ | $\alpha_{abs}$ of fossil fuel BC in the Aethalometer model | - | (6) |
| $\alpha_{bb}$ | $\alpha_{abs}$ of biomass-burning BC in the Aethalometer model | - | (6) |

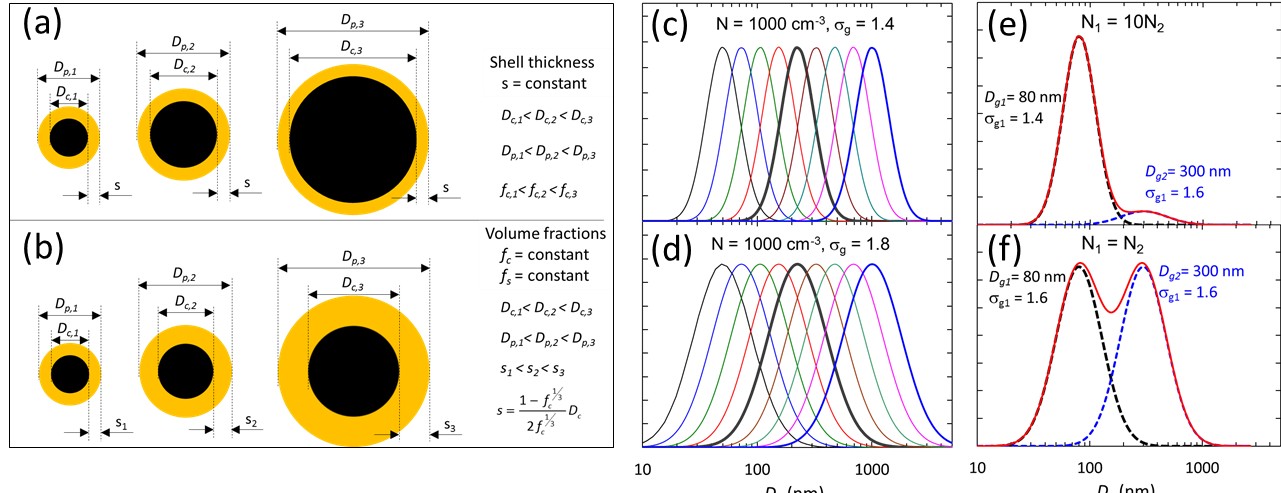

Figure 1. Examples of particles and size distributions used in the simulations: a) particles with a BC core

coated with a constant shell thickness s, b) particles with constant BC core and shell volume fractions $f_c$

and $f_s$, c) unimodal narrow size distributions with the geometric standard deviation of $\sigma_g$ = 1.4, d)

unimodal wide size distributions with $\sigma_g$ = 1.8, e) bimodal size distributions with a dominating Aitken

mode, f) bimodal size distributions with equal-sized Aitken and accumulation modes.

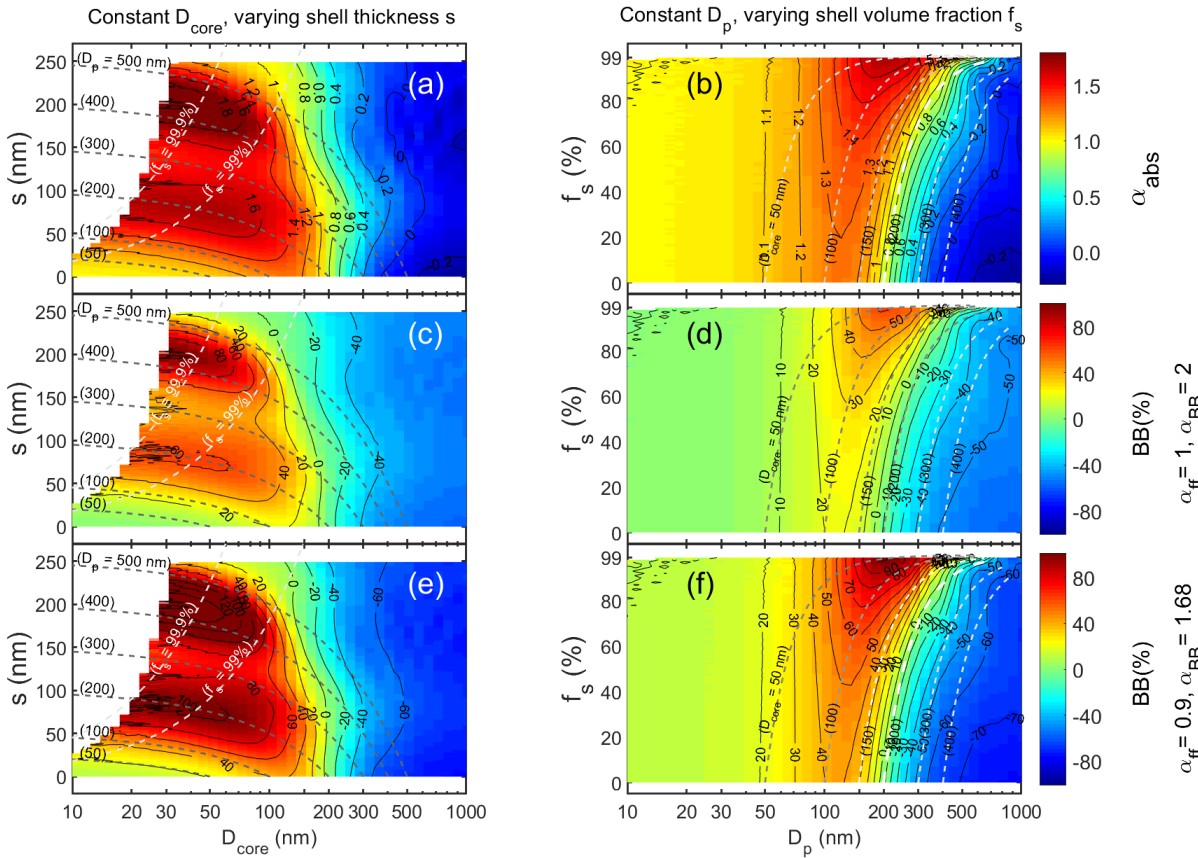

Figure 2. Absorption Ångström exponent ($\alpha_{abs}$) and the from it calculated fraction of biomass-burning BC for single coated particles as a function of (in -a, c, and e) BC core diameter ($D_{core}$) and shell thickness (s) and (b, d, and f) as a function of particle diameter ($D_p = D_{core}+2s$) and shell volume fraction $f_s$ in the range $f_s \leq 99$ %. In a), c) and e) the dark dashed lines show the $D_{core}$ and s of particles that have the same $D_p$ – written in parentheses – at all shell thicknesses and the light dashed line show the shell thicknesses that correspond to $f_s = 99\%$ and 99.9 %. In b), d) and f) the dashed lines show the $D_p$ and $f_s$ of particles that have the same $D_{core}$ – written in parentheses – at all shell volume fractions. The color bars are common for a and b, c and d, and e and f.

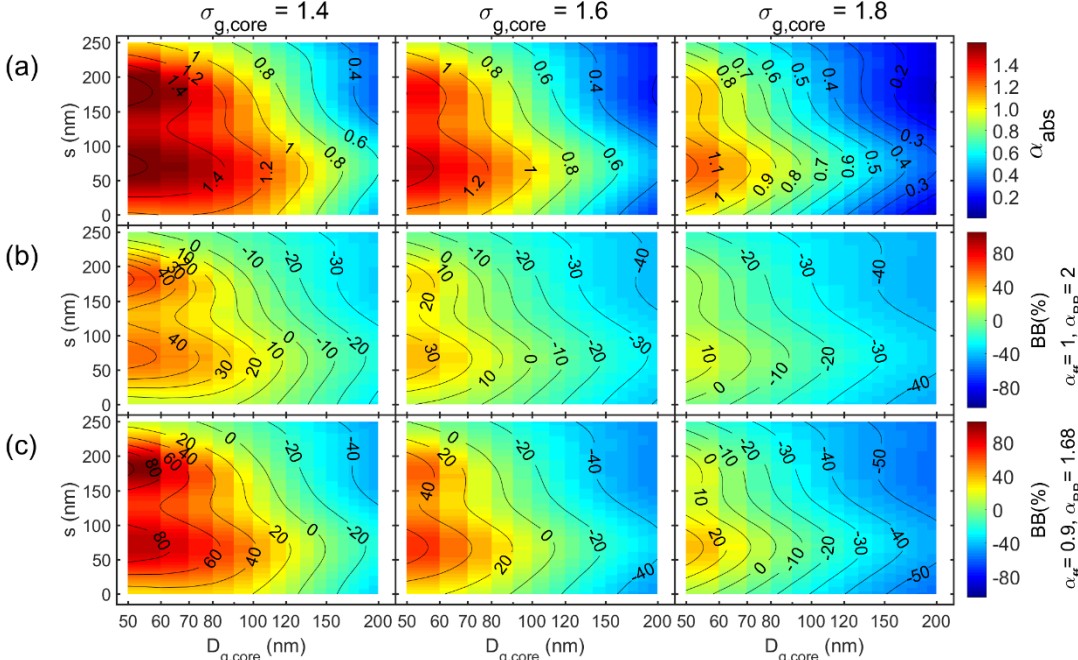

Figure 3. Unimodal particle size distributions with a size-independent shell thickness (s) for three widths

of the core size distributions: $\sigma_{g,core}$ = 1.4, 1.6 and 1.8.  a) absorption Ångström exponent ($\alpha_{abs}$) and the

from it calculated fraction of biomass-burning BC (BB(%)) with the Aethalometer model constants of b)

$\alpha_{ff}$ = 1, $\alpha_{bb}$ = 2  and c) $\alpha_{ff}$ = 0.9, $\alpha_{bb}$ = 1.68 vs. the geometric mean diameter of the core ($D_{g,core}$).

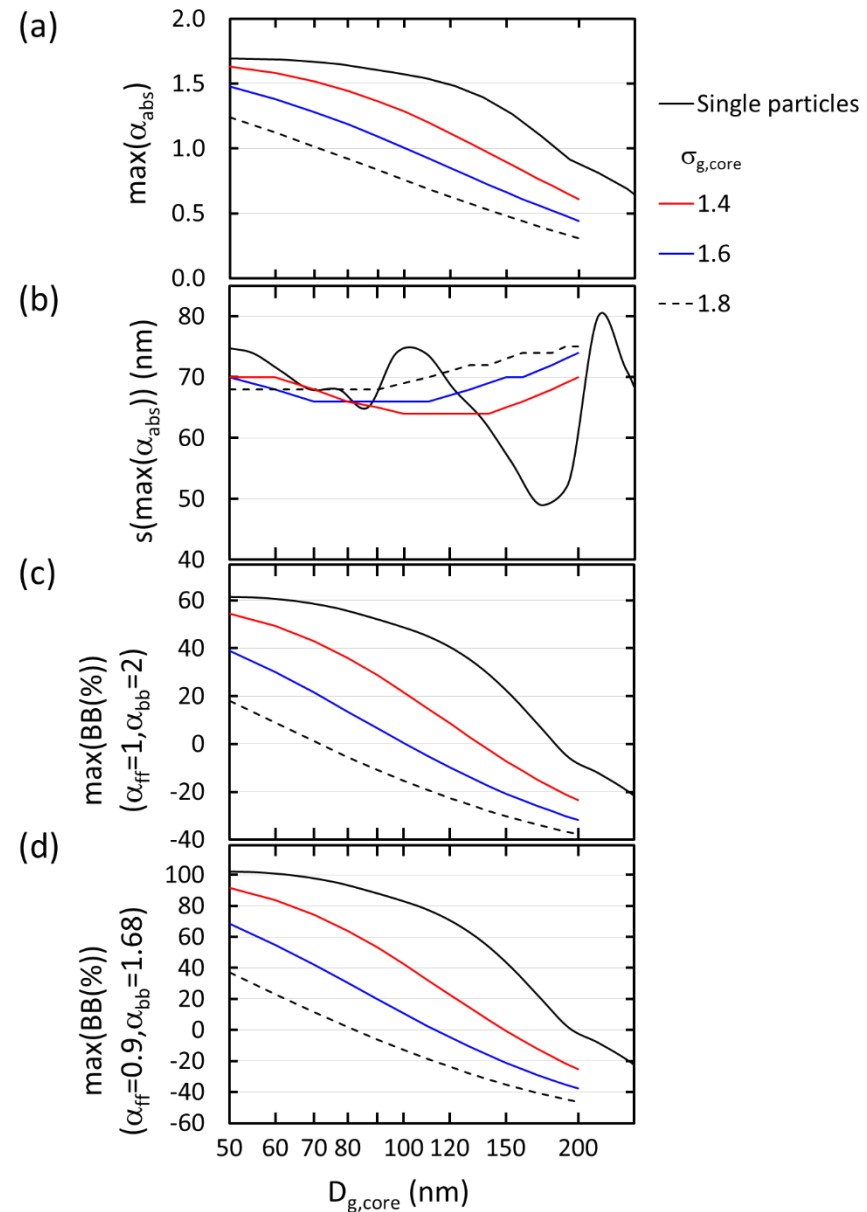

Figure 4. Size distribution dependence of the first maximum of $\alpha_{abs}$ when a size-independent shell grows on a BC core: a) the first maximum value of $\alpha_{abs}$, b) the shell thickness at the maximum $\alpha_{abs}$, c) maximum biomass-burning fraction with the Aethalometer model constants $\alpha_{ff} = 1$ and $\alpha_{bb} = 2$, and d) maximum biomass-burning fraction with the Aethalometer model constants $\alpha_{ff} = 0.9$ and $\alpha_{bb} = 1.68$ as a function of the BC core geometric mean diameter ($D_{g,core}$) and geometric standard deviation ($\sigma_{g,core}$) .

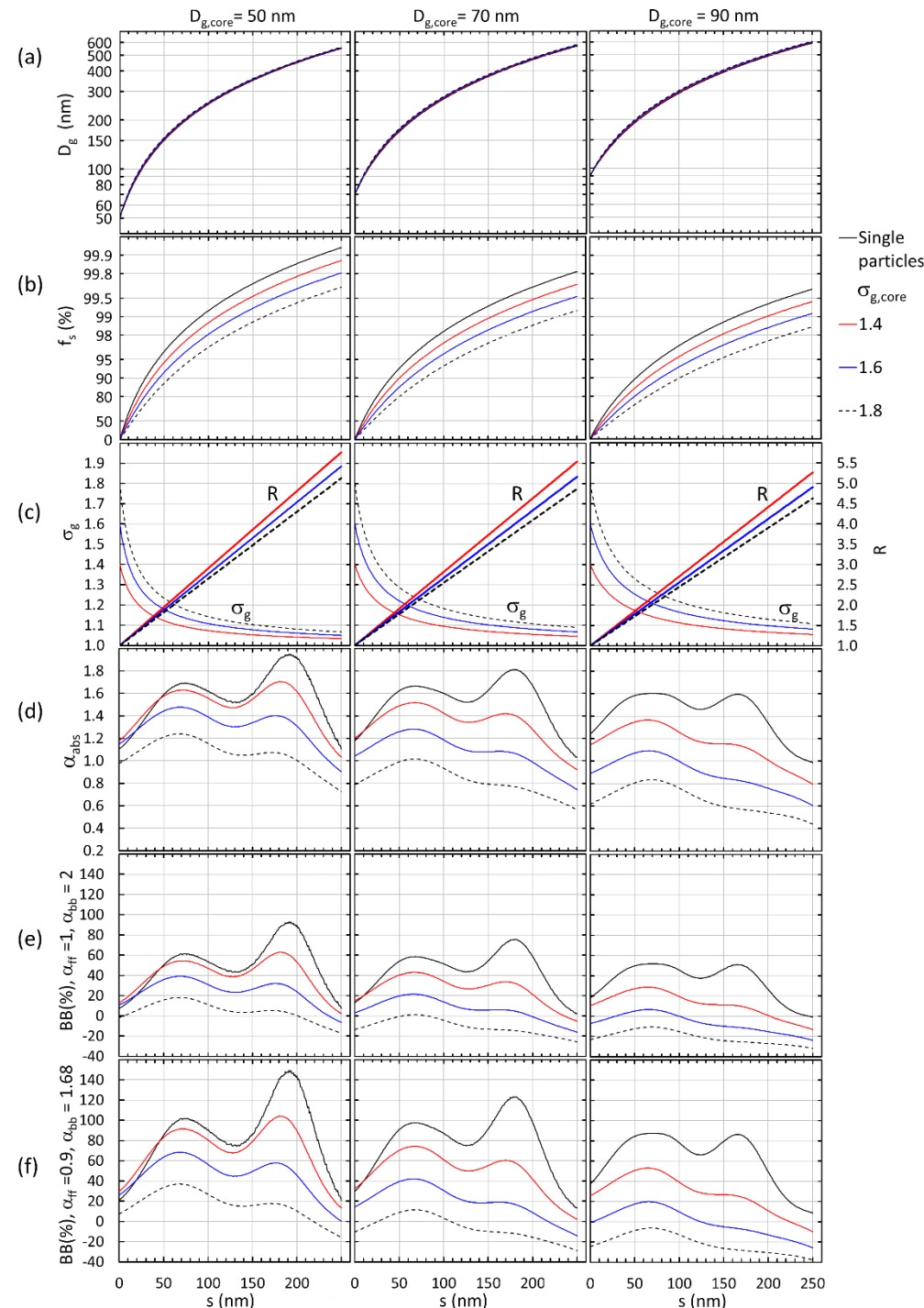

Figure 5. Examples of the growth of a non-size-dependent scattering shell on BC core size distributions with $D_{g,core}$ = 50 nm,

70 nm and 90 nm and on single BC particles. a) Geometric mean diameter, b) shell volume fraction, c) geometric standard

deviation and $D_p$-to-$D_{core}$ ratio (R), d) absorption Ångström exponent, e) BB(%) with the Aethalometer model constants $\alpha_{ff}$ =

1 and $\alpha_{bb}$ = 2, and d) biomass-burning fraction with the Aethalometer model constants $\alpha_{ff}$ = 0.9 and $\alpha_{bb}$ = 1.68 as a function

shell thickness s.

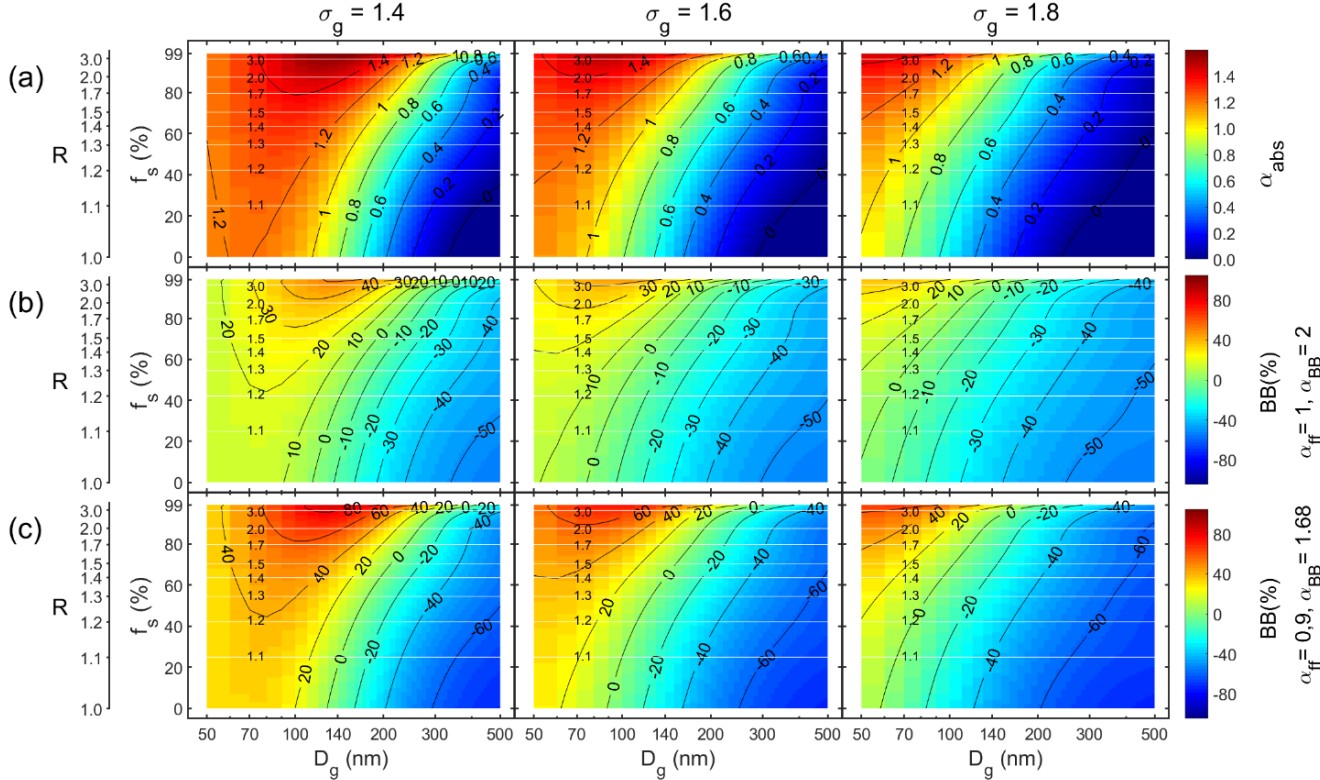

Figure 6. Unimodal particle size distributions with size-independent shell volume fractions $f_s$ and three

widths of the size distributions: $\sigma_g$ = 1.4, 1.6 and 1.8.  a) absorption Ångström exponent ($\alpha_{abs}$) and the

from it calculated fraction of biomass-burning BC (BB(%)) with the Aethalometer model constants of b)

$\alpha_{ff}$ = 1, $\alpha_{bb}$ = 2  and c) $\alpha_{ff}$ = 0.9, $\alpha_{bb}$ = 1.68 vs. the geometric mean diameter of the whole size distribution

($D_g$). The white horizontal grid lines show constant $D_p$-to-$D_{core}$ ratios (= R).

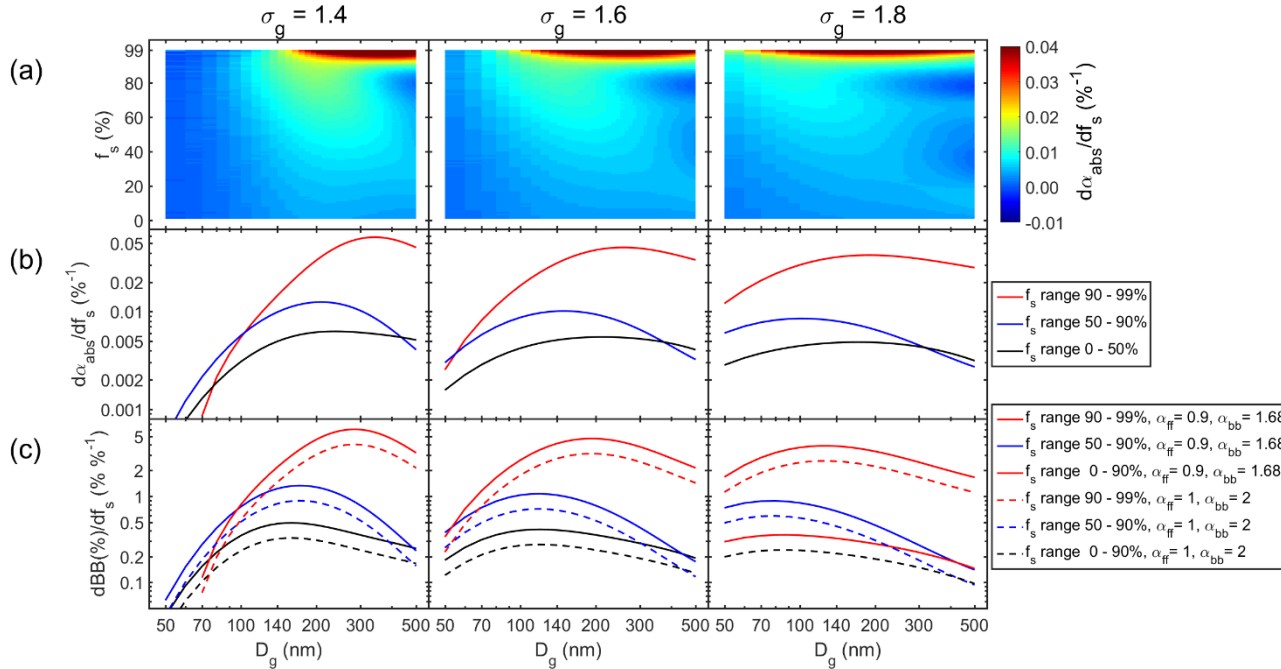

Figure 7. Size-dependent sensitivity of $\alpha_{abs}$ and BB(%) to variations of the shell volume fraction $f_s$. a) $\alpha_{abs}$ sensitivity in the whole $f_s$ range of 1 - 99%,  b) average $\alpha_{abs}$ sensitivity in three $f_s$ ranges, and (c) average BB(%) sensitivities in three $f_s$ ranges.

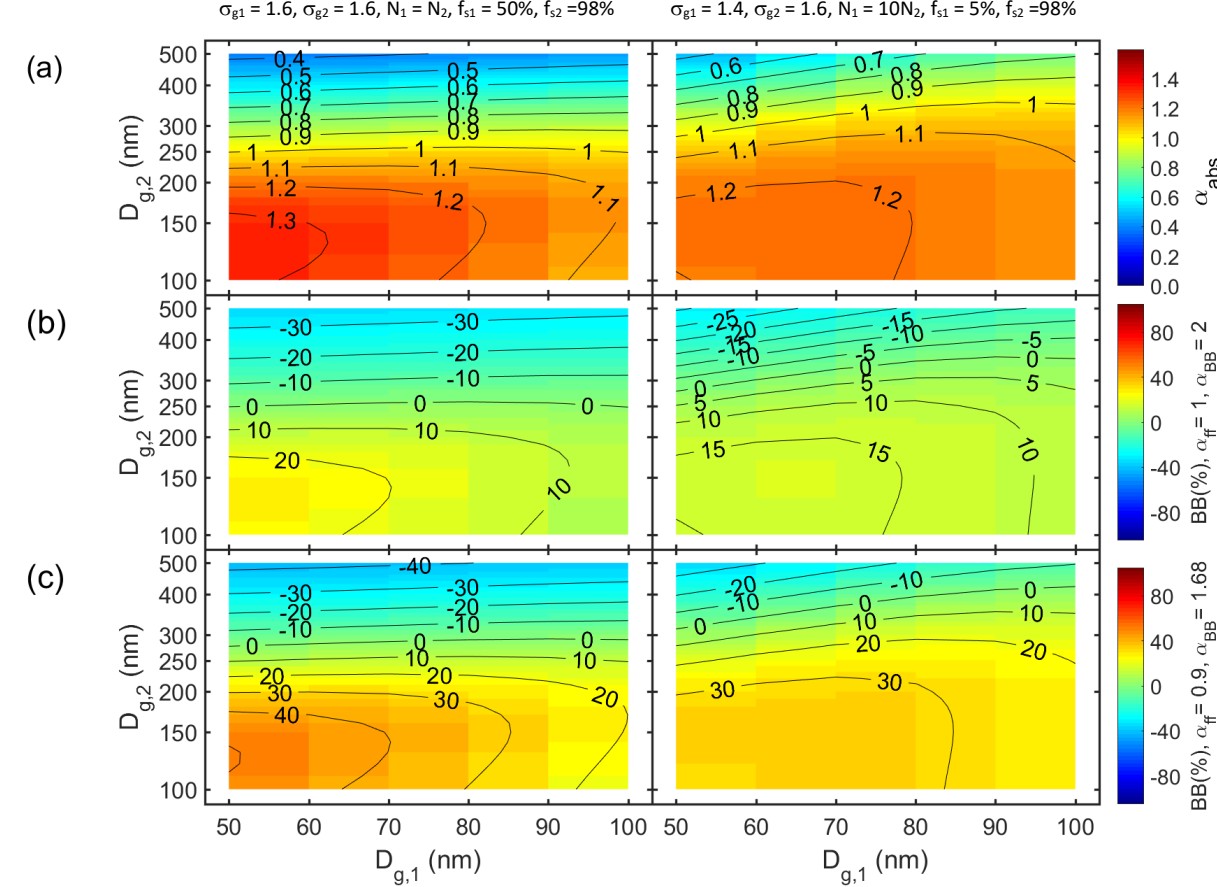

Figure 9. Bimodal particle size distributions with size-independent shell volume fractions $f_s$ in two modes as a function of geometric mean diameters of mode 1 ($D_{g1}$) and mode 2 ($D_{g2}$). a) absorption Ångström exponent ($\alpha_{abs}$) and the from it calculated fraction of biomass-burning BC (BB(%)) with the Aethalometer model constants of b) $\alpha_{ff} = 1$, $\alpha_{bb} = 2$ and c) $\alpha_{ff} = 0.9$, $\alpha_{bb} = 1.68$. The widths , the relative number of particles in the two modes and the shell volume fractions of the two modes on the left column: $\sigma_{g1} = 1.6$, $\sigma_{g2} = 1.6$, $N_1 = N_2$, $f_{s1} = 50\%$, $f_{s2} = 98\%$ and on the right column: $\sigma_{g1} = 1.4$, $\sigma_{g2} = 1.6$, $N_1 = 10N_2$, $f_{s1} = 5\%$, $f_{s2} = 98\%$.