# Peer review of "Modeled source apportionment of black carbon particles coated with a light-scattering shell"

_Atmospheric Measurement Techniques, 2020_

## Referee Comment (RC1) · Anonymous Referee #1 · 6 Jan 2021

The author simulated the absorption Ångström exponents of black carbon aerosol ensembles with varied particle morphologies and mixing states using the core-shell Mie model. The shapes of size distribution were presented for indicating the particle sizes of the entire BC-containing aerosol ensembles. The mixing states of coated BC particles are modeled using two morphologies: 1) particles with a BC core coated with a constant non-BC shell thickness; and 2) particles with the constant volume fractions of BC core and non-BC shell. Biomass-burning fractions were also calculated for pure and coated BC particles mixing with non-absorbing ammonium sulfate. This paper showed that the narrow size distributions result in higher absorption Ångström exponents and biomass-burning fractions than the wide size distributions. Moreover, the sensitivity of absorption Ångström exponents and biomass-burning fractions to variations in shell volume fractions is the highest for accumulation mode particles. These results are useful for this field, however, some technological issues are not clear in current manuscript and the advantages of these presented models compared to the current models should be highlighted. While you revise the paper, please take the following into consideration.

1. The method: absorption Ångström exponents of BC aerosol ensembles are calculated using two morphological models. The diversities of BC optical calculations between these two models are suggested to be investigated in this study. For a single BC aerosol ensemble, it can be described by these two models with the same volume-equivalent BC core size, non-BC shell size or the other parameters (as a single core-shell model). The same number-weighted Dp-to-Dcore ratio may also be a good option. The current comparison in Figure 2 showed the trend rather than the diversity. Moreover, the better model can also be supported by the observations.

2. In Section 3.2, the first and second maximum of BC absorption Ångström exponents is presented when a size-independent shell grows on a BC core. This point is useful, however, it seems like that the volume-equivalent BC core size and non-BC shell size of the cases are different. This presented variation was corresponding to the BC-containing particles with larger BC core size and smaller BC volume fraction. The situations may also be reproduced by those single particles with the same volume-equivalent BC core sizes and non-BC shell sizes.

3. Figure 7, the absorption properties of the single particles generally showed the similar non-monotonous variations with growing particle sizes. The assumption of the same BC volume fractions for all particles may also be simplified by the single core-shell model with the volume-equivalent BC core sizes and non-BC shell sizes.

4. The absorption Ångström exponent is an important indicator for the particle sizes and mixing states of black carbon aerosols. The simulations of the absorption Ångström exponent between 470nm and 950nm can be validated by the measure-

ments of AE33. Moreover, the other wavelength couples can also be simulated and compared to the observations. For example, the absorption Ångström exponent between 440nm and 870nm at all AERONET sites (Schuster et al., 2016).

5. Please check the acronym. For example, Line 19 of Page 3, Dc and Dcore may be the same. Define all the acronym in a list if possible, because they are confusing.

6. Please check the writing errors in current manuscript. For example, Line 19 of Page 6, 'thicnesses' may be 'thicknesses'.

Reference Schuster G L, Dubovik O, Arola A, et al. Remote sensing of soot carbon-Part 2: understanding the absorption Ångström exponent. Atmos Chem Phys 2016; 16(3): 1587-1602.

---

## Referee Comment (RC2) · Anonymous Referee #2 · 24 Jan 2021

Using the Core-Shell Mie theory, this study examines the absorption Ångström exponent (AAE) of BC particles with scattering coating (i.e., ammonium sulfate) by for different size and mode configurations. Then, the contributions of fossil fuel (FF) emissions and biomass burning (BB) to equivalent BC (eBC) in terms of biomass-burning contribution (BB%) are calculated from the simulated AAEs by using Aethalometer model. With these analysis, the author aims to study the potential uncertainties in the widely-used Aethalometer model for source appointment of eBCs. The article is well written, results are clearly described and discussed. I have a major concern related to study approach and experiment design.

If I understand correctly, the simulation experiments are for BC particles coated by ammonium sulfate to represent solely the fossil fuel aerosol type. In other words, the

simulated AAEs are for aerosols without biomass burning components, or BB% = 0. As a result, the calculated BB% that deviate from 0 would indicate uncertainty in source appointment. This is an important and basic experiment setting for the entire analysis and should be clearly stated in the article.

As such, there is a mismatch between the performed analysis and research goal that needs to be justified. As stated in article, the goal of this study is to evaluate uncertainties in the Aethalometer model for source appointment of eBCs. And I would expect to see some simulation experiments for BrC (in addition to FF). However, the entire analysis is for FF only and is unable to represent the case for presence of BrC. Therefore, the analysis in my opinion is incomplete, which primarily addressed the uncertainty for the assumption of AAE = 1.0 (or 0.9) for fossil fuel but not for the assumption of AAE = 2.0 (or 1.68) for BB component.

—————————————————————

---

## Author Comment (AC1) · 20 Mar 2021

**Detailed replies to Anonymous Referee #1**

**1. The method: absorption Ångström exponents of BC aerosol ensembles are calculated using two morphological models. The diversities of BC optical calculations between these two models are suggested to be investigated in this study. For a single BC aerosol ensemble, it can be described by these two models with the same volume-equivalent BC core size, non-BC shell size or the other parameters (as a single core-shell model). The same number-weighted Dp-to-Dcore ratio may also be a good option. The current comparison in Figure 2 showed the trend rather than the diversity. Moreover, the better model can also be supported by the observations.**

Thank you for this suggestion, it made me think through the equations. Before thinking I changed the code to make the simulations by using constant $D_p$-to-$D_{core}$ (= R) ratios for the whole size distribution and varied R from 1 to 4. But when I plotted the results they looked the same – with some y-axis scaling – as the results of the constant shell volume fraction ($f_s$) simulation presented in the discussion paper. I wondered why but a very simple calculation shows that they also should be. I added this text and equations (2) and (3) to the paper:

The ratio of the coated particle diameter to the core diameter is an often used metric for presenting the coating of particles. $R$, $f_c$ and $f_s$ can be calculated from each other as

$$R = \frac{D_p}{D_{core}} = \left(\frac{1}{f_c}\right)^{\frac{1}{3}} = \left(\frac{1}{1-f_s}\right)^{\frac{1}{3}} \qquad (2)$$

The number-weighted $D_p$-to-$D_{core}$ ratio is calculated from

$$R_{N(D_p)} = \frac{\sum N_i R_i}{N_{tot}} = \frac{\sum N_i \left(D_{p,i}/D_{core,i}\right)}{N_{tot}} \qquad (3)$$

where $N_i$ and $R_i$ are the number concentration and $D_p$-to-$D_{core}$ ratio of the particle diameter $D_{p,i}$, respectively. If $f_s$ is independent of particle size – which is the assumption used in some of the simulations below – equation (3) simplifies to $R_{N(Dp)} = R$.

In addition I added a secondary y axis to Fig. 6, showing R values side-by-side with the original y axis that shows the $f_s$ values. I also added white grid lines showing constant R values. I could just well have added a similar secondary axis to Fig 2 but it is already now so heavily loaded with information that I considered additional information to make it too busy. I also added the following text explaining the secondary y axis in Fig. (6) in Section 3.3:

Note that from Eq. (3) it follows that the assumption of a constant $f_s$ means that alse the $D_p$-to-$D_{core}$ ratio R is constant and that the $f_s$ range of 0 to 99% corresponds to the R range of 1 to 4.6. Figure 6 therefore has two y axes, one showing the $f_s$ and the other the corresponding R values.

**2. In Section 3.2, the first and second maximum of BC absorption Ångström exponents is presented when a size-independent shell grows on a BC core. This point is useful, however, it seems like that the volume-equivalent BC core size and non-BC shell size of the cases are different. This presented variation was corresponding to the BC-containing particles with larger BC core size and smaller BC volume fraction. The situations may also be reproduced by those single particles with the same volume equivalent BC core sizes and non-BC shell sizes.**

I start from this statement: "…**it seems like that the volume-equivalent *BC core size and non-BC shell size of the cases are different***." I am confused and simply don't understand this statement. In section I first present Fig. 3 where the y axis is the shell thickness (s) and the x axis is the geometric mean diameter of the core ($D_{g,core}$) in three cases: geometric standard deviation $\sigma_g$ of 1.4, 1.6 and 1.8. For sure both s and $D_{g,core}$ are the same in all these cases, only $\sigma_g$ is different. Then in Fig. 4a I present the $\alpha_{abs}$ at the first maximum $\alpha_{abs}$ seen on the color-scaled plot in Fig. 3a as a function of $D_{g,core}$ for the size distributions and as a function of $D_{core}$ in Fig. 2a for single particles. Again, in all these cases $D_{g,core}$ and $D_{core}$ are the same, only $\sigma_g$ is different. In the last figure of the section, Fig. 5, the effects of the size-independent growth on BC core size distributions with $D_{g,core}$= 50, 70, and 90 nm and on single partices with $D_{core}$ = 50, 70 and 90 nm as a function of s are shown. The subfigures Fig. 5d, 5e, and 5f can be considered as "vertical slices" from Figs 3a, 3b, 3c, 2a, 2c, and 2e. Again, in all these cases $D_{g,core}$ and $D_{core}$ are the same, only $\sigma_g$ is different. So, what does this statement "…***BC core size and non-BC shell***

*size of the cases are different.*" mean? And I don't understand what I am expected to do. Should I correct something?

I continue with this statement: "***This presented variation was corresponding to the BC-containing particles with larger BC core size and smaller BC volume fraction.***" What does this statement mean? In Fig. 3 I present the variations of $\alpha_{abs}$, and BB fractions for size distributions with $D_{g,core}$ varying from 50 to 200 nm and s varying from 0 to 250 nm. So not just large cores and small BC volume fractions. I am again confused and don't know what I am expected to do.

The last sentence of question 2. reads "***The situations may also be reproduced by those single particles with the same volume equivalent BC core sizes and non-BC shell sizes.***" This is also a comment I don't understand. In section 3.2 there are three figures, Fig. 3, 4, and 5. Fig. 3 shows the properties of size distributions that were presented for single particles in Fig. 2. In Figs. 4 and 5 there are the lines for both size distributions and single particles. So, haven't I done already what seems to be suggested in the statement? What exactly is missing?

**3. Figure 7, the absorption properties of the single particles generally showed the similar non-monotonous variations with growing particle sizes. The assumption of the same BC volume fractions for all particles may also be simplified by the single coreshell model with the volume-equivalent BC core sizes and non-BC shell sizes.**

Fig. 7 presents size-dependent sensitivity of $\alpha_{abs}$ and BB(%) to variations of the shell volume fraction $f_s$ in size distributions. They are the derivatives $d\alpha_{abs}/df_s$ and $dBB(\%)/df_s$ of the $\alpha_{abs}$ and BB(%) values presented in Fig. 6. Yes, the reviewer's comment "… ***the absorption properties of the single particles generally showed the similar non-monotonous variations with growing particle sizes.***" is correct, the variations of $\alpha_{abs}$ and BB(%) shown for single particles in Fig. 2 are similar and they could have been included in Fig. 7 as a fourth column of subfigures. I considered this is not necessary and the reviewer did not require it. Fig. 7 already now quite clearly delivers the main messages: the sensitivity increases when the shell grows and the sensitivity depends on the width of the size distribution and on the geometric mean diameter of the size distribution.

The second sentence "***The assumption of the same BC volume fractions for all particles may also be simplified by the single coreshell model with the volume-equivalent BC core sizes and non-BC shell sizes.***" is again a bit confusing. Do you wish I change the figure to derivatives of $d\alpha_{abs}/dR$ and $dBB(\%)/dR$ where R is the $D_p$-to-$D_{core}$ ratio R? If so, I don't uderstand how this would simplify the figure. Or what is meant with this statement.

**4. The absorption Ångström exponent is an important indicator for the particle sizes and mixing states of black carbon aerosols. The simulations of the absorption Ångström exponent between 470nm and 950nm can be validated by the measure ments of AE33. Moreover, the other wavelength couples can also be simulated and compared to the observations. For example, the absorption Ångström exponent between 440nm and 870nm at all AERONET sites (Schuster et al., 2016).**

This is a good statement. I've been dealing with filter-based absorption photometers and I am not really familiar with AOD data processing and undeliberately didn't even consider other wavelength pairs. But now that you suggested this I repeated the calculations for the wavelength pair 440/870, present the results in an appendix and wrote this additional text to the paper:

==In analyses of aerosol optical depth data from the AERONET network $\alpha_{abs}$ is often calculated for the wavelength pair 440 nm and 870 nm (Russell et al., 2010; Schuster et al., 2016). To evaluate the applicability of the simulations of the present work to AERONET data analyses $\sigma_{ap}$ was calculated also for these wavelengths and the respective $\alpha_{abs}$ was calculated from them. There are size-dependent differences between $\alpha_{abs}(470/950)$ and $\alpha_{abs}(440/870)$ but they are not big, see the supplement, Figs. S1 and S2, so it may safely be concluded that the results to be presented below are valid also for the AERONET data.==

I did not add any corresponding figures to the main paper, however, since the main point of the paper is to evaluate the Aethalometer model that uses the Aethalometer wavelengths.

**5. Please check the acronym. For example, Line 19 of Page 3, Dc and Dcore may be the same. Define all the acronym in a list if possible, because they are confusing.**

> This is a good observation and suggestion. I tried to find and remove all confusing symbols and acronyms and added a table according to the suggestion: Table 1. Nomenclature

**6. Please check the writing errors in current manuscript. For example, Line 19 of Page 6, 'thicnesses' may be 'thicknesses'.**

> I have now used MS Word spell check and read through the text. I cannot swear it is error free but tried my best now.

**Reference Schuster G L, Dubovik O, Arola A, et al. Remote sensing of soot carbon - Part 2: understanding the absorption Ångström exponent. Atmos Chem Phys 2016; 16(3): 1587-1602.**

> Done

---

## Author Comment (AC2) · 20 Mar 2021

**Detailed replies to Anonymous Referee #2**

**If I understand correctly, the simulation experiments are for BC particles coated by ammonium sulfate to represent solely the fossil fuel aerosol type. In other words, the simulated AAEs are for aerosols without biomass burning components, or BB% = 0. As a result, the calculated BB% that deviate from 0 would indicate uncertainty in source appointment. This is an important and basic experiment setting for the entire analysis and should be clearly stated in the article.**

> This is a good point. I added these sentences at the end of the introduction:
> ==To state this more clearly, it is assumed that there is only one type of BC particles that can be called fossil fuel BC in the Aethalometer model terminology. Consequently, any deviations from biomass-burning fraction of BB% = 0 indicate uncertainty in the source appointment.==

**As such, there is a mismatch between the performed analysis and research goal that needs to be justified. As stated in article, the goal of this study is to evaluate uncertainties in the Aethalometer model for source appointment of eBCs. And I would expect to see some simulation experiments for BrC (in addition to FF). However, the entire analysis is for FF only and is unable to represent the case for presence of BrC. Therefore, the analysis in my opinion is incomplete, which primarily addressed the uncertainty for the assumption of AAE = 1.0 (or 0.9) for fossil fuel but not for the assumption of AAE = 2.0 (or 1.68) for BB component.**

> There is definitely not even a slightest mismatch between the goal and the analysis.
>
> It is not the goal of the paper to find out whether some pair of $\alpha_{ff}$ and $\alpha_{bb}$ is better than the other. The whole AE model is based on the use of three absorption Ångström exponents: the measured $\alpha_{abs}$ and the preset constants $\alpha_{ff}$ and $\alpha_{bb}$. Different values for these constants have been presented and are in a wide use. In numerous field studies $\alpha_{abs}$ is measured and these constants are used for source apportionment without any supporting data, especially on BC core size distribution or coating thickness. Here I just use these two well-known $\alpha_{ff}$ and $\alpha_{bb}$ pairs and show how large the uncertainties may become just for these two pairs even if BC particles were coated by purely scattering material, not to claim that either of these two pairs is "better".
>
> The goal is not at all to find a good pair. The analysis shows that no constant values are good since BC particle size distributions are not constant, neither their mean diameter nor their coating. They all vary dynamically in the atmosphere. It really does not matter which values of these constants are selected for showing that any constant values will undoubtedly lead to small or large uncertainties of both the BB and FF fractions if no information on the size of the core or the thickness of the shell is available, even if only purely scattering material is coating BC cores.
>
> That is the whole point of the paper.
>
> There is no reason to make simulations using imaginary refractive indices deviating from zero for the coating material. That would be contrary to the goal of the paper. Even the title of the paper reads
>
> *Modeled source apportionment of black carbon particles **coated with a light-scattering shell***
>
> It is another story when measurements are used to constrain model results. Then it makes sense to vary the full complex refractive indices of the core and shell(s).